# Depth-Bounds for Neural Networks
# via the Braid Arrangement

**Moritz Grillo**
Max Planck Institute for Mathematics in the Sciences
`moritz.grillo@mis.mpg.de`

**Christoph Hertrich**
University of Technology Nuremberg
`christoph.hertrich@utn.de`

**Georg Loho**
Freie Universität Berlin
University of Twente
`georg.loho@math.fu-berlin.de`

## Abstract

We contribute towards resolving the open question of how many hidden layers are required in ReLU networks for exactly representing all continuous and piecewise linear functions on $\mathbb{R}^d$. While the question has been resolved in special cases, the best known lower bound in general is still 2. We focus on neural networks that are compatible with certain polyhedral complexes, more precisely with the braid fan. For such neural networks, we prove a non-constant lower bound of $\Omega(\log \log d)$ hidden layers required to exactly represent the maximum of $d$ numbers. Additionally, we provide a combinatorial proof that neural networks satisfying this assumption require three hidden layers to compute the maximum of 5 numbers; this had only been verified with an excessive computation so far. Finally, we show that a natural generalization of the best known upper bound to maxout networks is not tight, by demonstrating that a rank-3 maxout layer followed by a rank-2 maxout layer is sufficient to represent the maximum of 7 numbers.

## 1   Introduction

Among the various types of neural networks, ReLU networks have become particularly prominent [Glorot et al., 2011, Goodfellow et al., 2016]. For a thorough theoretical understanding of such neural networks, it is important to analyze which classes of functions we can represent with which depth. Classical universal approximation theorems [Cybenko, 1989, Hornik, 1991] ensure that just one hidden layer can approximate any continuous function on a bounded domain with arbitrary precision. However, establishing an analogous result for *exact* representations remains an open question and is the subject of ongoing research [Arora et al., 2018, Hertrich et al., 2023, Haase et al., 2023, Valerdi, 2024, Averkov et al., 2025].

While in practical settings approximate representations are often sufficient, studying the exact piecewise linear structure of neural network representations enabled deep connections between neural networks and fields like tropical and polyhedral geometry [Huchette et al., 2023]. These connections, in turn, are important for algorithmic tasks like neural network training [Arora et al., 2018, Goel et al., 2021, Khalife and Basu, 2022, Froese et al., 2022, Froese and Hertrich, 2023, Bertschinger et al., 2023] and verification [Li et al., 2019, Katz et al., 2017, Froese et al., 2025b,a, Stargalla et al., 2025], including understanding the computational complexity of the respective tasks.

Arora et al. [2018] initiate the study of exact representations by showing that the class of functions exactly representable by ReLU networks is the class of continuous piecewise linear (CPWL) functions.

39th Conference on Neural Information Processing Systems (NeurIPS 2025).

Specifically, they demonstrate that every CPWL function defined on $\mathbb{R}^d$ can be represented by a ReLU network with $\lceil \log_2(d+1) \rceil$ hidden layers. This result is based on Wang and Sun [2005], who reduce the representation of a general CPWL function to the representation of maxima of $d+1$ affine terms. By computing pairwise maxima in each layer, such a maximum of $d+1$ terms can be computed with logarithmic depth overall in the manner of a binary tree. Very recently, Bakaev et al. [2025b] improved the upper bound by proving that every CPWL function can be represented with $\lceil \log_3(d-1) \rceil + 1$ hidden layers. Their results refute the conjecture of Hertrich et al. [2023] that $\lceil \log_2(d+1) \rceil$ hidden layers are indeed necessary to compute all CPWL functions.

Based on the result by Wang and Sun [2005], Hertrich et al. [2023] deduced that it suffices to determine the minimum depth representation of the maximum function. While it is easy to show that $\max\{0, x_1, x_2\}$ cannot be represented with one hidden layer [Mukherjee and Basu, 2017], Bakaev et al. [2025b] showed that two hidden layers are sufficient to represent $\max\{0, x_1, x_2, x_3, x_4\}$. However, it remains open if there exists a CPWL function on $\mathbb{R}^d$ that really needs logarithmic many hidden layers to be represented. In particular, it is already open whether there is a function that needs more than two hidden layers to be represented.

Understanding depth lower bounds is important for clarifying the potential advantages of architectural choices. In particular, proving depth lower bounds on computing the max function helps formally explain why elements like max-pooling layers are powerful and cannot be easily replaced by shallow stacks of standard ReLU layers, regardless of their width.

In order to identify tractable special cases to prove lower bounds on the necessary number of hidden layers to compute the max function, two approaches have been pursued so far. The first restricts the possible *breakpoints* of all neurons in a network computing $x \mapsto \max\{0, x_1, \ldots, x_d\}$. A breakpoint of a neuron is an input for which the function computed by the neuron is non-differentiable. A neural network is called $\mathcal{B}_d^0$-*conforming* if breakpoints only appear where the ordering of some pair of coordinates changes (i.e., all breakpoints lie on hyperplanes $x_i = x_j$ or $x_i = 0$). While $\mathcal{B}_d^0$-conforming networks can compute the max function with $\lceil \log_2(d+1) \rceil$ hidden layers, Hertrich et al. [2023] show that 2 hidden layers are insufficient to compute the function $\max\{0, x_1, x_2, x_3, x_4\}$, using a computational proof via a mixed integer programming formulation of the problem. The second approach restricts the weights of the network. Averkov et al. [2025] show that, if all weights are $N$-ary fractions, the max function can only be represented by neural network with depth $\Omega(\frac{\log d}{\log \log N})$ by extending an approach of Haase et al. [2023]. Furthermore, Bakaev et al. [2025a] proved lower bounds for the case when some or all weights are restricted to be nonnegative. To the best of our knowledge, the two approaches of restricting either the breakpoints or the weights are incomparable.

**Our contributions** We follow the approach from Hertrich et al. [2023] and prove lower bounds on $\mathcal{B}_d^0$-conforming networks. On one hand, following Hertrich et al. [2023], we believe that understanding $\mathcal{B}_d^0$-conforming networks might also shed light on the expressivity of general networks, for example, by studying different underlying fans instead of focusing on the braid fan as an intermediate step. On the other hand, $\mathcal{B}_d^0$-conforming also appears in Brandenburg et al. [2025] and Froese et al. [2025b] due to the connection to submodular functions and graphs.

In Section 4 we prove for $d = 2^{2^\ell - 1}$ that the function $x \mapsto \max\{0, x_1, \ldots, x_d\}$ is not representable with a $\mathcal{B}_d^0$-conforming ReLU network with $\ell$ hidden layers. This means that depth $\Omega(\log \log d)$ is necessary for computing all CPWL functions, yielding the first conditional non-constant lower bound without restricting the weights of the neural networks.

To prove our results, the first observation is that the set of functions that are representable by a $\mathcal{B}_d^0$-conforming network forms a finite-dimensional vector space (Proposition 2.2). While one would like to identify subspaces of this vector space representable with a certain number of layers, taking the maximum of two functions does not behave well with the structure of linear subspaces. To remedy this, we identify a suitable sequence of subspaces $\mathcal{F}_\mathcal{L}(k)$ for $k = 1, 2, \ldots$ that can be controlled through an inductive construction. These auxiliary subspaces arise from the correspondence between $\mathcal{B}_d^0$-conforming functions and set functions. This allows us to employ the combinatorial structure of the collection of all subsets of a finite ground set. This is also reflected in the structure of the breakpoints of $\mathcal{B}_d^0$-conforming functions. Hence, we are able to show that applying a rank-2-maxout-layer to functions in $\mathcal{F}_\mathcal{L}(k)$ yields a function in $\mathcal{F}_\mathcal{L}(k^2 + k)$. Iterating this argument yields the desired bounds.

In Section 5, we focus on the case $d = 4$. We provide a combinatorial proof of the result of Hertrich et al. [2023] showing that the function $x \mapsto \max\{0, x_1, x_2, x_3, x_4\}$ is not representable by a $\mathcal{B}_d^0$-conforming ReLU network with two hidden layers.

Finally, in Section 6, we study maxout networks as natural generalization of ReLU networks. A straightforward generalization of the upper bound of Arora et al. [2018] shows that $\mathcal{B}_d^0$-conforming maxout network with ranks $r_i$ in the hidden layers $i = 1, \ldots, \ell$ can compute the maximum of $\prod_{i=1}^{\ell} r_i$ numbers. We prove that this upper bound is not tight: a maxout network with one rank-3 layer and one rank-2 layer can compute the maximum of 7 numbers, that is, the function $x \mapsto \max\{0, x_1, \ldots, x_6\}$.

**Further Related Work**   In light of the prominent role of the max function for neural network expressivity, Safran et al. [2024] studied efficient neural network approximations of the max function.

In an extensive line of research, tradeoffs between depth and size of neural networks have been explored, demonstrating that deep networks can be exponentially more compact than shallow ones [Montúfar et al., 2014, Telgarsky, 2016, Eldan and Shamir, 2016, Arora et al., 2018, Ergen and Grillo, 2024]. While most of these works also involve lower bounds on the depth, they are usually proven under assumptions on the width. In contrast, we aim towards proving lower bounds on the depth for unrestricted width. The opposite perspective, namely studying bounds on the size of neural networks irrespective of the depth, has been subject to some research using methods from combinatorial optimization [Hertrich and Skutella, 2023, Hertrich and Sering, 2024, Hertrich and Loho, 2024].

One of the crucial techniques in expressivity questions lies in connections to tropical geometry via Newton polytopes of functions computed by neural networks. This was initiated by Zhang et al. [2018], see also Maragos et al. [2021], and subsequently used to understand decision boundaries, bounds on the depth, size, or number of linear pieces, and approximation capabilities [Montúfar et al., 2022, Misiakos et al., 2022, Haase et al., 2023, Brandenburg et al., 2024, Valerdi, 2024, Hertrich and Loho, 2024].

## 2   Preliminaries

In Appendix A, the reader can find an overview of the notation used in the paper and in Appendix B detailed proofs of all the statements.

**Polyhedra**   We review basic definitions from polyhedral geometry; see Schrijver [1986], Ziegler [2012] for more details.

A *polyhedron $P$* is the intersection of finitely many closed halfspaces and a *polytope* is a bounded polyhedron. A hyperplane *supports $P$* if it bounds a closed halfspace containing $P$, and any intersection of $P$ with such a supporting hyperplane yields a *face $F$* of $P$. A face is a *proper face* if $F \subsetneq P$ and $F \neq \emptyset$ and inclusion-maximal proper faces are referred to as *facets*. A *(polyhedral) cone $C \subseteq \mathbb{R}^n$* is a polyhedron such that $\lambda u + \mu v \in C$ for every $u, v \in C$ and $\lambda, \mu \in \mathbb{R}_{\geq 0}$. A cone is *pointed* if it does not contain a line. A cone $C$ is *simplicial*, if there are linearly independent vectors $v_1, \ldots, v_k \in \mathbb{R}^n$ such that $C = \{\sum_{i=1}^k \lambda_i v_i \mid \lambda_i \geq 0\}$.

A *polyhedral complex $\mathcal{P}$* is a finite collection of polyhedra such that (i) $\emptyset \in \mathcal{P}$, (ii) if $P \in \mathcal{P}$ then all faces of $P$ are in $\mathcal{P}$, and (iii) if $P, P' \in \mathcal{P}$, then $P \cap P'$ is a face both of $P$ and $P'$. A polyhedral *fan* is a polyhedral complex where all polyhedra are cones. The *lineality space* of a polyhedron $P$ is defined as $\{v \in \mathbb{R}^d \mid x + v \in P \text{ for all } x \in P\}$. The lineality space of a polyhedral complex $\mathcal{P}$ is the lineality space of one (and therefore all) $P \in \mathcal{P}$.

**Neural networks and CPWL functions**   A continuous function $f \colon \mathbb{R}^n \to \mathbb{R}$ is called *continuous and piecewise linear* (CPWL), if there exists a polyhedral complex $\mathcal{P}$ such that the restriction of $f$ to each full-dimensional polyhedron $P \in \mathcal{P}^n$ is an affine function. If this condition is satisfied, we say that $f$ and $\mathcal{P}$ are *compatible* with each other. We denote the set of all CPWL functions from $\mathbb{R}^d$ to $\mathbb{R}$ by $\mathrm{CPWL}_d$.

For a number of hidden layers $\ell \geq 0$, a *neural network* with *rectified linear unit* (ReLU) activation is defined by a sequence of $\ell + 1$ affine maps $T_i \colon \mathbb{R}^{n_{i-1}} \to \mathbb{R}^{n_i}$, $i \in [\ell + 1]$. We assume that $n_0 = d$ and $n_{\ell+1} = 1$. If $\sigma$ denotes the function that computes the ReLU function $x \mapsto \max\{x, 0\}$

in each component, the neural network is said to compute the CPWL function $f\colon \mathbb{R}^d \to \mathbb{R}$ given by $f = T_{\ell+1} \circ \sigma \circ T_\ell \circ \sigma \circ \cdots \circ \sigma \circ T_1$.

A *rank-r-maxout layer* is defined by $r$ affine maps $T^{(q)}\colon \mathbb{R}^d \to \mathbb{R}^n$ for $q \in [r]$ and computes the function $x \mapsto (\max\{(T^{(1)}x)_j, \ldots, (T^{(r)}x)_j\})_{j\in[n]}$. For a number of hidden layers $\ell \geq 0$ and a rank vector $\mathbf{r} = (r_1, \ldots, r_\ell) \in \mathbb{N}^\ell$, a *rank-$\mathbf{r}$-maxout neural network* is defined by maxout layers $f_i\colon \mathbb{R}^{n_{i-1}} \to \mathbb{R}^{n_i}$ of rank $r_i$ for $i \in [\ell]$ respectively and an affine transformation $T_{out}\colon \mathbb{R}^{n_\ell} \to \mathbb{R}$. The rank-$\mathbf{r}$-maxout neural network computes the function $f\colon \mathbb{R}^d \to \mathbb{R}$ given by $f = T_{out} \circ f_\ell \circ \cdots \circ f_1$. Let $\mathcal{M}_d^{\mathbf{r}}$ be the set of functions representable by a rank-$\mathbf{r}$-maxout neural network with input dimension $d$. Moreover, let $\mathcal{M}_d^2(\ell)$ be the set of functions representable with networks with $\ell$ rank-2-maxout layers.

### The braid arrangement and set functions

**Definition 2.1.** *The* braid arrangement *in $\mathbb{R}^d$ is the hyperplane arrangement consisting of the $\binom{d}{2}$ hyperplanes $x_i = x_j$, with $1 \leq i < j \leq d$. The* braid fan $\mathcal{B}_d$ *is the polyhedral fan induced by the braid arrangement.*

Sometimes we will also refer to the fan given by the $\binom{d+1}{2}$ hyperplanes $x_i = x_j$ and $x_i = 0$ for $1 \leq i < j \leq d$, which we denote by $\mathcal{B}_d^0$.

We summarize the properties of the braid fan that are relevant for this work. For more details see Stanley [2007]. The $k$-dimensional cones of $\mathcal{B}_d$ are given by

$$\{\mathrm{cone}(\mathbb{1}_{S_1}, \ldots, \mathbb{1}_{S_k}) + \mathrm{span}(\mathbb{1}_{[d]}) \mid \emptyset \subsetneq S_1 \subsetneq S_2 \subsetneq \cdots \subsetneq S_k \subsetneq [d]\},$$

where $\mathbb{1}_S = \sum_{i \in S} e_i$. The braid fan has $\mathrm{span}(\mathbb{1}_{[d]})$ as lineality space. Dividing out the lineality space of $\mathcal{B}_d$ yields $\mathcal{B}_{d-1}^0$. See Figure 1a for an illustration of $\mathcal{B}_d^0$.

Using the specific structure of the cones of $\mathcal{B}_d$ in terms of subsets of $[d]$ allows to relate the vector space $\mathcal{V}_{\mathcal{B}_d}$ of CPWL functions compatible with the braid fan $\mathcal{B}_d$ with the vector space of set functions $\mathcal{F}_d := \mathbb{R}^{2^{[d]}}$: restricting to the values on $\{\mathbb{1}_S\}_{S\subseteq[d]}$ yields a vector space isomorphism $\Phi\colon \mathcal{V}_{\mathcal{B}_d} \to \mathcal{F}_d$ whose inverse map is given by interpolating the values on $\{\mathbb{1}_S\}_{S\subseteq[d]}$ to the interior of the cones of the braid fan. Detailed proofs of all statements can be found in Appendix B.

**Proposition 2.2.** *The linear map $\Phi\colon \mathcal{V}_{\mathcal{B}_d} \to \mathcal{F}_d$ given by $F(S) := \Phi(f)(S) = f(\mathbb{1}_S)$ is an isomorphism.*

This implies that $\mathcal{V}_{\mathcal{B}_d}$ has dimension $2^d$. Another basis for $\mathcal{V}_{\mathcal{B}_d}$ is given by $\{\sigma_M \mid M \in 2^{[d]}\}$, where the function $\sigma_M\colon \mathbb{R}^d \to \mathbb{R}$ is defined by $\sigma_M(x) = \max_{i \in M} x_i$ [Danilov and Koshevoy, 2000, Jochemko and Ravichandran, 2022]. We have the following strict containment of linear subspaces:

$$\mathcal{V}_{\mathcal{B}_d}(0) \subsetneq \mathcal{V}_{\mathcal{B}_d}(1) \subsetneq \ldots \subsetneq \mathcal{V}_{\mathcal{B}_d}(d) = \mathcal{V}_{\mathcal{B}_d}$$

where $\mathcal{V}_{\mathcal{B}_d}(k) := \mathrm{span}\{\sigma_M \mid M \subseteq [d], |M| \leq k\}$. In order to describe the linear subspaces $\Phi(\mathcal{V}_{\mathcal{B}_d}(k))$, we now describe the isomorphism $\Phi$ with respect to the basis $\{\sigma_M \mid M \in 2^{[d]}\}$.

Let $X$ and $Y$ be finite sets such that $X \subseteq Y$, then the interval $[X, Y] := \{S \subseteq [Y] \mid X \subseteq S\}$ is a *Boolean lattice* with the partial order given by inclusion. The *rank* of $[X, Y]$ is given by $|Y \setminus X|$. Sometimes we also write $x_1 \cdots x_n$ for the set $\{x_1, \ldots, x_n\} \in \mathcal{L}$ and $\overline{x_1 \cdots x_n}$ for the set $X \cup (Y \setminus \{x_1, \ldots, x_n\})$. For a Boolean lattice $\mathcal{L} = [X, Y]$ of rank $n$, the *rank function* $r\colon \mathcal{L} \to [n]_0$ is given by $r(S) = |S| - |X|$ and $r(S)$ is called the *rank* of $S$. Moreover, we define the *levels* of a Boolean lattice by $\mathcal{L}_i := r^{-1}(i)$ and introduce the notation $\mathcal{L}_{\leq i} := \bigcup_{j \leq i} \mathcal{L}_j$ for the set of elements whose rank is bounded by $i$. For $S, T \in \mathcal{L}$ with $S \subseteq T$, we call $[S, T]$ a *sublattice* of $\mathcal{L}$ and define the vector $\alpha_{S,T} \in \mathbb{R}^{\mathcal{L}}$ by $\alpha_{S,T} := \sum_{S \subseteq Q \subseteq T} (-1)^{r(Q)-r(S)} \mathbb{1}_Q$. The set $\mathcal{F}_{\mathcal{L}} := (\mathbb{R}^{\mathcal{L}})^*$ of real-valued functions on $\mathcal{L}$ is a vector space, and for any fixed $S, T \in \mathcal{L}$, the map $F \mapsto \langle \alpha_{S,T}, F \rangle$ is a linear functional of $\mathcal{F}_{\mathcal{L}}$. Furthermore, let

$$\mathbb{R}^{\mathcal{L}}(k) = \mathrm{span}\{\alpha_{S,T} \mid S, T \in \mathcal{L}, S \subseteq T \text{ such that } r(T) - r(S) = k + 1\}$$

and $\mathcal{F}_{\mathcal{L}}(k) := (\mathbb{R}^{\mathcal{L}}(k))^{\perp} = \{F \in \mathcal{F}_{\mathcal{L}} \mid \langle \alpha_{S,T}, F \rangle = 0 \text{ for all } \alpha_{S,T} \in \mathbb{R}^{\mathcal{L}}(k)\}$ be a linear subspace of $\mathcal{F}_{\mathcal{L}}$. To simplify notation, we also set $\mathcal{F}_d(k) := \mathcal{F}_{2^{[d]}}(k)$.

**Proposition 2.3.** *The isomorphism $\Phi \colon \mathcal{V}_{\mathcal{B}_d} \to \mathcal{F}_d$ maps the function $f = \sum_{M \subseteq [d]} \lambda_M \cdot \sigma_M$ to the set function defined by $F(S) := \sum_{\substack{M \subseteq [d] \\ M \cap S \neq \emptyset}} \lambda_M \cdot \sigma_M$. The inverse $\Phi^{-1} \colon \mathcal{F}_d \to \mathcal{V}_{\mathcal{B}_d}$ of $\Phi$ is given by the Möbius inversion formula $F \mapsto \sum_{M \subseteq [d]} -\langle \alpha_{[d] \setminus M, [d]}, F \rangle$. In particular, it holds that $\Phi(\mathcal{V}_{\mathcal{B}_d}(k)) = \mathcal{F}_d(k)$ for all $k \leq d$ and $\dim(\mathcal{F}_d(k)) = \dim(\mathcal{V}_{\mathcal{B}_d}(k)) = \sum_{i=1}^{k} \binom{d}{i}$. See also Figure 1b for an illustration of Proposition 2.3.*

## 3 Neural networks conforming with the braid fan

For a polyhedral complex $\mathcal{P}$, we call a maxout neural network $\mathcal{P}$-*conforming*, if the functions at all neurons are compatible with $\mathcal{P}$. By this we mean that for all $i \in [\ell]$ and all coordinates $j$ of the codomain of $f_i$, the function $\pi_j \circ f_i \circ \ldots \circ f_1$ is compatible with $\mathcal{P}$, where $\pi_j$ is the projection on the coordinate $j$. We denote by $\mathcal{M}_{\mathcal{P}}^{\mathbf{r}}$ the set of all functions representable by $\mathcal{P}$-conforming rank-$\mathbf{r}$-maxout networks. For the remainder of this article, we only consider the cases $\mathcal{M}_{\mathcal{B}_d}^{\mathbf{r}}$ and $\mathcal{M}_{\mathcal{B}_d^0}^{\mathbf{r}}$

**Lemma 3.1.** *The function $x \mapsto \max\{0, x_1, \ldots, x_{d-1}\}$ can be represented by a $\mathcal{B}_{d-1}^0$-conforming rank-$\mathbf{r}$-maxout network if and only if the function $x \mapsto \max\{x_1, \ldots, x_d\}$ can be represented by a $\mathcal{B}_d$-conforming rank-$\mathbf{r}$-maxout network.*

By computing $r_i$ maxima in each layer, we can compute the basis functions of $\mathcal{V}_{\mathcal{B}_d}(\prod_{i=1}^{\ell} r_i)$ with a $\mathcal{B}_d$-conforming rank-$\mathbf{r}$-maxout network.

**Proposition 3.2.** *For any rank vector $\mathbf{r} \in \mathbb{N}^\ell$, it holds that all functions in $\mathcal{V}_{\mathcal{B}_d}(\prod_{i=1}^{\ell} r_i)$ are representable by a $\mathcal{B}_d$-conforming rank-$\mathbf{r}$-maxout network.*

Most of the paper is concerned with proving that $\mathcal{M}_{\mathcal{B}_d}^r$ is contained in certain subspaces of $\mathcal{V}_{\mathcal{B}_d}$. Let $\mathcal{F}_{\mathcal{L}}^r = \bigoplus_{i \in [r]} \mathcal{F}_{\mathcal{L}}$ be the $r$-fold direct sum of $\mathcal{F}_{\mathcal{L}}$ with itself. In order to model the application of the rank-$r$-maxout activation function for a set function under the isomorphism $\Phi$, we define for $(F_1, \ldots, F_r) \in \mathcal{F}_{\mathcal{L}}^r$ the function $\max\{F_1, \ldots, F_r\} \in \mathcal{F}_{\mathcal{L}}$ given by $\max\{F_1, \ldots, F_r\}(S) = \max\{F_1(S), \ldots, F_r(S)\}$.

For $f_1, \ldots, f_r \in \mathcal{V}_{\mathcal{B}_d}$, the function $\max\{f_1, \ldots, f_r\}$ is $\mathcal{B}_d$-compatible if taking the maximum does not create breakpoints that do not lie on the braid arrangement, that is, on every cone $C$ of the braid arrangement, it holds that $\max\{f_1, \ldots, f_r\} = f_q$ for a $q \in [r]$. Next, we aim to model the compatibility with the braid arrangement for set functions. We call a tuple $(F_1, \ldots, F_r) \in \mathcal{F}_{\mathcal{L}}^r$ *conforming* if for every chain $\emptyset = S_0 \subsetneq S_1 \subsetneq \ldots \subsetneq S_n \subseteq [n]$ there is a $j \in [r]$ such that $F_j(S_i) = \max\{F_1, \ldots, F_r\}(S_i)$ for all $i \in [n]_0$. Then, the set $\mathcal{C}_{\mathcal{L}}^r \subseteq \mathcal{F}_{\mathcal{L}}^r$ of conforming tuples are exactly those tuples of CPWL functions such that applying the maxout activation function yields a function that is still compatible with the braid fan as stated in the next lemma. Again, to simplify notation, we also set $\mathcal{C}_d^r := \mathcal{C}_{2^{[d]}}^r$.

**Lemma 3.3.** *For $(F_1, \ldots, F_r) \in (\mathcal{F}_d)^r$, the function $\max\{\Phi^{-1}(F_1), \ldots, \Phi^{-1}(F_r)\}$ is $\mathcal{B}_d$-conforming if and only if $(F_1, \ldots, F_r) \in \mathcal{C}_d^r$. In this case,*

$$\max\{\Phi^{-1}(F_1), \ldots, \Phi^{-1}(F_r)\} = \Phi^{-1}(\max\{F_1, \ldots, F_r\})$$

The statement ensures that taking the maximum of the set functions is the same as taking the maximum of the piecewise-linear functions exactly for compatible tuples.

## 4 Doubly-logarithmic lower bound

In this section, we prove that for any number of layers $\ell \in \mathbb{N}$, the function $\max\{0, x_1, \ldots, x_{2^{2^\ell}-1}\}$ is not computable by a $\mathcal{B}_d^0$-conforming rank-2-maxout neural network (or equivalently ReLU neural network) with $\ell$ hidden layers. Due to the equivalence of $\mathcal{B}_d$ and $\mathcal{B}_d^0$, we will prove that $\mathcal{M}_{\mathcal{B}_d}^2(\ell) \subseteq \mathcal{V}_{\mathcal{B}_d}(2^{2^\ell-1})$ for $d \geq 2^{2^\ell-1} + 1$.

First, we define an operation $\mathcal{A}$ on subspaces of $\mathcal{V}_{\mathcal{B}_d}$ that describes rank-2-maxout layers that maintain compatibility with $\mathcal{B}_d$. For any subspace $U \subseteq \mathcal{V}_{\mathcal{B}_d}$, let $\mathcal{A}(U) \subseteq \mathcal{V}_{\mathcal{B}_d}$ be the subspace containing all

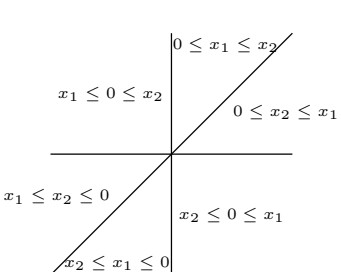

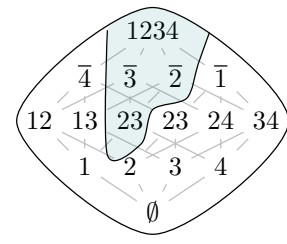

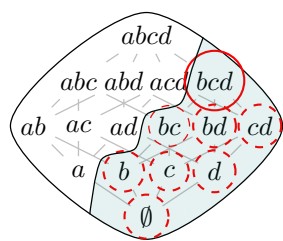

(a) The braid arrangement $\mathcal{B}_2^0$.

(b) Illustration of Proposition 2.3. The coefficient of the function $\sigma_{\{2,4\}}$ in the linear combination for $F$ is given by $-\langle \alpha_{13,1234}, F \rangle$.

(c) Illustration of Lemma 4.5. If $F \in \mathcal{F}_{\mathcal{L}}(2) \cap \mathcal{C}_{\mathcal{L}}$ and $F(\{b,c,d\}) < 0$, then there is a $S \subsetneq \{b,c,d\}$ with $F(S) < 0$ since $\langle \alpha_{\emptyset,bcd}, F \rangle = 0$.

Figure 1

the functions computable by a $\mathcal{B}_d$-conforming rank 2-maxout layer that takes functions from $U$ as input. Formally,

$$\mathcal{A}(U) = \text{span}\{\max\{f_1, f_2\} \mid f_1, f_2 \in U, \max\{f_1, f_2\} \in \mathcal{V}_{\mathcal{B}_d}\}.$$

Clearly, $\mathcal{A}(U_1)$ is a subspace of $\mathcal{A}(U_2)$ whenever $U_1$ is a subspace of $U_2$. We recursively define $\mathcal{A}^\ell(U) = \mathcal{A}(\mathcal{A}^{\ell-1}(U))$. This recursive definition allows to describe the set of $\mathcal{B}_d$-conforming network with $\ell$ rank-2-maxout layers $\mathcal{M}_{\mathcal{B}_d}^2(\ell)$.

**Lemma 4.1.** *It holds that (1) $\mathcal{M}_{\mathcal{B}_d}^2(1) = \mathcal{A}(\mathcal{V}_{\mathcal{B}_d}(1)) = \mathcal{V}_{\mathcal{B}_d}(2)$, and (2) for all $\ell \in \mathbb{N}$, $\mathcal{M}_{\mathcal{B}_d}^2(\ell) = \mathcal{A}(\mathcal{M}_{\mathcal{B}_d}^2(\ell-1)) = \mathcal{A}^\ell(\mathcal{V}_{\mathcal{B}_d}(1))$.*

Since it holds that $\max\{f_1, f_2\} = \max\{0, f_1 - f_2\} + f_2$, we can assume wlog that one of the functions is the zero map, as stated in the following lemma.

**Lemma 4.2.** *It holds that $\mathcal{A}(U) = \text{span}\{\max\{0, f\} \mid f \in U, \max\{0, f\} \in \mathcal{V}_{\mathcal{B}_d}\}$.*

To prove that $\mathcal{M}_{\mathcal{B}_d}^2(\ell) = \mathcal{A}^\ell(\mathcal{V}_{\mathcal{B}_d}(1))$ is a proper subspace of $\mathcal{V}_{\mathcal{B}_d}$ for $d \geq 2^{2^\ell - 1} + 1$, we perform a layerwise analysis and inductively bound $n_k$ depending on $k$ such that $\mathcal{A}(\mathcal{V}_{\mathcal{B}_d}(k)) \subseteq \mathcal{V}_{\mathcal{B}_d}(n_k)$ for all $k \in \mathbb{N}$. In this attempt, we translate this task to the setting of set functions on Boolean lattices using the isomorphism $\Phi$. Recall that the pairs $(F_1, F_2) \in \mathcal{C}_{\mathcal{L}}^2$ are precisely the functions such that the maximum of the corresponding CPWL functions $f_1$ and $f_2$ is still compatible with $\mathcal{B}_d$. Moreover, it is easy to observe, that the pair $(0, F) \in \mathcal{F}_{\mathcal{L}}^2$ is conforming if and only if $F$ is contained in the set

$$\mathcal{C}_{\mathcal{L}} := \{F \in \mathcal{F}_{\mathcal{L}} \mid F(S) \text{ and } F(T) \text{ do not have opposite signs for } S \subseteq T\}.$$

Again, to simplify notation, we also set $\mathcal{C}_d := \mathcal{C}_{2^{[d]}}$ and use the notation $F^+ = \max\{0, F\}$. By slightly overloading notation, for any subspace $U \subseteq \mathcal{F}_{\mathcal{L}}$, let $\mathcal{A}(U) = \text{span}\{F^+ \mid F \in U \cap \mathcal{C}_{\mathcal{L}}\}$. Lemma 3.3 justifies this notation and allows us to carry out the argumentation to the world of set functions on Boolean lattices, as we conclude in the following lemma.

**Lemma 4.3.** *It holds that $\mathcal{A}(\Phi(U)) = \Phi(\mathcal{A}(U))$ for all subspaces $U \subseteq \mathcal{V}_{\mathcal{B}_d}$. In particular, for any lattice $\mathcal{L} = [X, Y]$, it holds that $\mathcal{A}(\mathcal{F}_{\mathcal{L}}(1)) = \mathcal{F}_{\mathcal{L}}(2)$.*

In the following, we prove that $\mathcal{A}(\mathcal{F}_{\mathcal{L}}(k)) \subseteq \mathcal{F}_{\mathcal{L}}(k^2 + k)$ by an induction on $k$ and Lemma 4.3 serves as the base case.

Next, we describe properties of the vector space $\mathbb{R}^{\mathcal{L}}$ that will be useful for the induction step. Every sublattice of $\mathcal{L}$ of rank $k+1$ is of the form $[S, S \cup T]$, where $S \cap T = \emptyset$ and $|T| = k + 1$. For any $T \subseteq Y \setminus X$, one can decompose $\mathcal{L} = [X, Y]$ into the sublattices $[S, S \cup T]$ for all $S \subseteq Y \setminus T$, resulting in the following lemma.

**Lemma 4.4.** *Let $\mathcal{L} = [X, Y]$ be a lattice of rank $n$. Then, (1) for every $T \subseteq Y \setminus X$, it holds that $\alpha_{X,Y} \in \text{span}\{\alpha_{S, S \cup T} \mid S \subseteq Y \setminus T\}$, and (2) for every $T \subseteq Y \setminus X$ with $|T| = k$, it holds that $\alpha_{S, S \cup T} - \alpha_{S', S' \cup T} \in \mathbb{R}^{\mathcal{L}}(k)$ for all $S, S' \in [X, Y \setminus T]$.*

See Figure 2 for a visualization of Lemma 4.4. Lemma 4.4 implies that it suffices to find a $T \subseteq Y$ such that $\langle \alpha_{S, S \cup T}, F^+ \rangle = 0$ for all $S \subseteq Y \setminus T$, in order to prove that $\mathcal{F}_{\mathcal{L}}(n-1)$. The idea of the

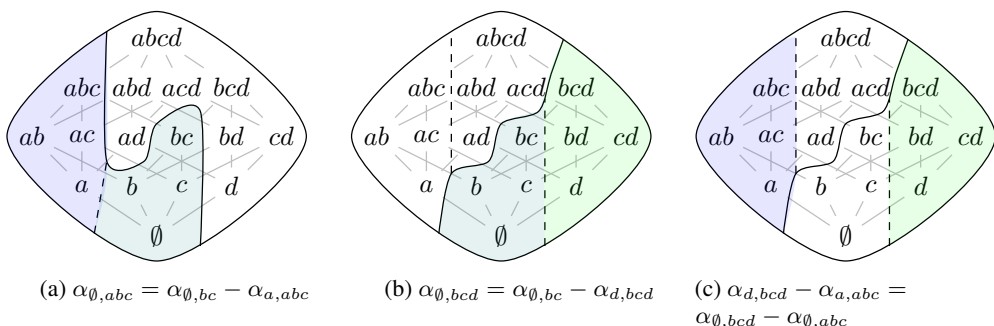

(a) $\alpha_{\emptyset,abc} = \alpha_{\emptyset,bc} - \alpha_{a,abc}$     (b) $\alpha_{\emptyset,bcd} = \alpha_{\emptyset,bc} - \alpha_{d,bcd}$     (c) $\alpha_{d,bcd} - \alpha_{a,abc} =$
$\alpha_{\emptyset,bcd} - \alpha_{\emptyset,abc}$

Figure 2: Illustration of Lemma 4.4. The solid line in Figure 2a, decomposes the lattice in $[\emptyset, abc] \cup [d, abcd]$, which implies that $\alpha_{\emptyset,abcd} = \alpha_{\emptyset,abc} - \alpha_{d,abcd}$. The dashed line further decomposes $[\emptyset, abc] = [\emptyset, bc] \cup [a, abc]$. The 3 figures illustrate that $\alpha_{S,S\cup\{b,c\}} - \alpha_{S',S'\cup\{b,c\}} \in \mathbb{R}^{\mathcal{L}}(2)$ for all $S, S' \subseteq \{a, d\}$.

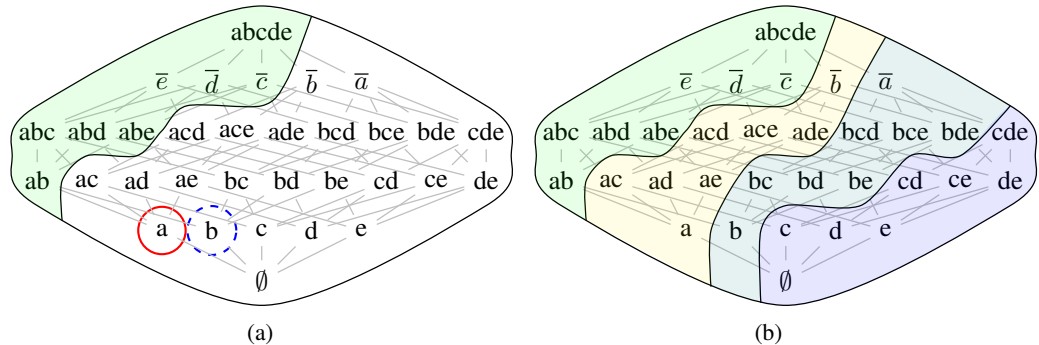

(a)                      (b)

Figure 3: An illustration of the induction step. Let $Y = \{a, b, c, d, e\}, X = \emptyset, \mathcal{L} = [X, Y]$ and $F \in \mathcal{F}_{\mathcal{L}}(2) \cap \mathcal{C}_{\mathcal{L}}$. If $F(a) < 0$ and $F(b) > 0$, then it follows that $F(R)$ for all $R \in [S, S \cup T]$ for $S = ab$ and $T = cde$ (Figure 3a). In particular, $F \in \mathcal{F}_{S,S\cup T}(1)$ and thus, by Lemma 4.4, it holds that $F \in \mathcal{F}_{S',S'\cup T}(1)$ for all $S' \subseteq Y \setminus T$.

Figure 3b shows the decomposition of the lattice $\mathcal{L} = [X, Y]$ for $T = \{c, d, e\}$ into the sublattices $[S, S \cup T]$ for all $S \subseteq Y \setminus T$. For every such sublattice we have that $F \in \mathcal{F}_{[S,S\cup T]}(1) \cap \mathcal{C}_{[S,S\cup T]}$ and thus by induction $\langle \alpha_{S,S\cup T}, F^+ \rangle = 0$.

induction step is to find a $T$ of cardinality at least $(k-1)^2 + (k-1) + 1$ such that $F \in \mathcal{F}_{[S,S\cup T]}(k-1)$ for all $S \subseteq Y \setminus T$. Then, applying the induction hypothesis to each sublattice $[S, S \cup T]$ yields $\langle \alpha_{S,S\cup T}, \overline{F}^+ \rangle = 0$ and hence $F^+ \in \mathcal{F}_{\mathcal{L}}(n-1)$.

If $F \in \mathcal{F}_{\mathcal{L}}(k)$, Lemma 4.4 implies that for any $T' \subseteq Y \setminus X$ of cardinality $k$, the value $\langle \alpha_{S',S'\cup T'}, F \rangle$ is independent of $S' \subseteq Y \setminus T'$. Hence, in this case, it suffices to find a $T$ such that $F \in \mathcal{F}_{[S,S\cup T]}(k-1)$ for only one $S \subseteq Y \setminus T$, since it is equivalent to $F \in \mathcal{F}_{[S,S\cup T]}(k-1)$ for all $S \subseteq Y \setminus T$.

Given $F \in \mathcal{F}_{\mathcal{L}}(k) \cap \mathcal{C}_{\mathcal{L}}$, it remains to find such $S$ and $T$. We define the *support* of $F \in \mathcal{F}_{\mathcal{L}}$ by $\mathrm{supp}(F) = \{S \in \mathcal{L} \mid F(S) \neq 0\}$ and the *positive and negative support* by $\mathrm{supp}^+(F) = \{S \in \mathcal{L} \mid F(S) > 0\}$ respectively $\mathrm{supp}^-(F) = \{S \in \mathcal{L} \mid F(S) < 0\}$. In particular, $F \in \mathcal{C}_{\mathcal{L}}$ implies that for $X^+ \in \mathrm{supp}^+(F)$ and $X^- \in \mathrm{supp}^-(F)$, it holds that $F(R) = 0$ for all $R \supseteq X^+ \cup X^-$.

Lemma 4.5 says that, given that the positive and negative support are not empty, we can always "push the elements $X^+$ and $X^-$ in the support down in the lattice", that is, we can find elements in the supports that are of relatively low rank. See Figure 1c for an illustration.

**Lemma 4.5.** *Let $\mathcal{L} = [X, Y]$ be a lattice of rank $n$. Let $F \in \mathcal{F}_{\mathcal{L}}(k) \cap \mathcal{C}_{\mathcal{L}}$ such that $F \ngeq 0$ and $F \nleq 0$. Then, there are $X^- \in \mathcal{L}_{\leq k} \cap \mathrm{supp}^-(F)$ and $X^+ \in \mathcal{L}_{\leq k} \cap \mathrm{supp}^+$ as well as $Y^- \in \mathcal{L}_{\geq n-k} \cap \mathrm{supp}^-(F)$ and $Y^+ \in \mathcal{L}_{\geq n-k} \cap \mathrm{supp}^+(F)$.*

Let $S = X^+ \cup X^-$, then $F \in \mathcal{C}_{\mathcal{L}}$ implies that for $T = Y \setminus S$, we have that $F(R) = 0$ for all $R \in [S, S \cup T]$. In particular, it holds that $F \in \mathcal{F}_{[S,S\cup T]}(k-1)$. Thus, by Lemma 4.4, if $F \in \mathcal{F}_{\mathcal{L}}(k)$,

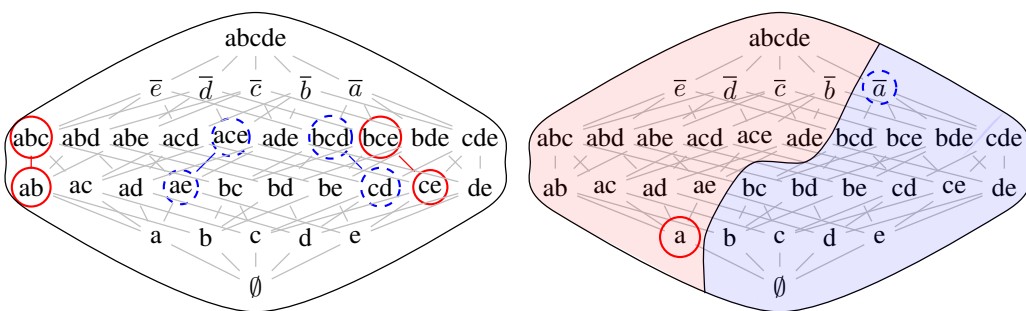

Figure 4: An illustration of Lemma 5.1 (left) and Lemma B.1 (right). If $\mathrm{supp}(F) \subseteq \mathcal{L}_2 \cup \mathcal{L}_3$, then we can match every $S \in \mathcal{L}_2$ with a $T \in \mathcal{L}_3$ such that $F(T) = F(S)$ which implies $\langle \alpha_{\emptyset, abcde}, F^+ \rangle = \sum_{S \in \mathcal{L}_2} F^+(S) - \sum_{T \in \mathcal{L}_3} F^+(T) = 0$. If $F(a) < 0$ and $F(bcde) > 0$, then it holds that $\langle \alpha_{\emptyset, abcde}, F \rangle = \langle \alpha_{\emptyset, bcde}, F \rangle = 0$.

it follows that $F \in \mathcal{F}_{[S', S' \cup T]}(k-1)$ for all $S' \subseteq Y \setminus T'$. Since $|S|$ is at most $2k$ it follows by counting that if $n \geq (k^2 + k + 1)$, the cardinality of $T$ is at least $(k-1)^2 + (k-1) + 1$. This allows to apply the inductions hypothesis to all sublattices $[S', S' \cup T]$ for $S' \subseteq Y \setminus T$, resulting in the following proposition. See also Figure 3b for an illustration of the induction.

**Proposition 4.6.** *For $k \in \mathbb{N}$, let $\mathcal{L} = [X, Y]$ be a lattice of rank $n \geq k^2 + k + 1$ and $F \in \mathcal{F}_\mathcal{L}(k) \cap \mathcal{C}_\mathcal{L}$. Then it holds that $\langle \alpha_{X,Y}, F^+ \rangle = 0$*

Applying Proposition 4.6 to every sublattice of rank $k^2 + k + 1$ allows to sharpen the bound.

**Proposition 4.7.** *Let $\mathcal{L}$ be a lattice and $k \in \mathbb{N}$, then it holds that $\mathcal{A}(\mathcal{F}_\mathcal{L}(k)) \subseteq \mathcal{F}_\mathcal{L}(k^2 + k)$.*

Translating this result back to the CPWL functions and applying the argument iteratively for a rank-2-maxout network, layer by layer, we obtain the following theorem.

**Theorem 4.8.** *For a number of layers $\ell \in \mathbb{N}$, it holds that $\mathcal{M}^2_{\mathcal{B}_d}(\ell) \subseteq \mathcal{V}_{\mathcal{B}_d}(2^{2^\ell - 1})$.*

**Corollary 4.9.** *The function $x \mapsto \{0, x_1, \ldots, x_{2^{2^\ell - 1}}\}$ is not computable by a $\mathcal{B}^0_d$-conforming ReLU neural network with $\ell$ hidden layers.*

## 5 Combinatorial proof for dimension four

In this section, we prove that the function $\max\{0, x_1, \ldots, x_4\}$ cannot be computed by a $\mathcal{B}^0_d$-conforming rank-$(2,2)$-maxout networks or equivalently ReLU neural networks with 2 hidden layers. This completely classfies the set of functions computable by $\mathcal{B}_d$-conforming ReLU neural networks with 2 hidden layers.

If $\mathcal{L}$ is a lattice of rank 5 and $F \in \mathcal{F}_\mathcal{L}(2) \cap \mathcal{C}_\mathcal{L}$, we know by Lemma 4.5, given that the supports of $F$ are not empty, that there are $X^+ \in \mathcal{L}_2 \cap \mathrm{supp}^+(F)$ and $X^- \in \mathcal{L}_2 \cap \mathrm{supp}^-(F)$. We first argue that in the special case of rank 5 we can even assume that there are $X^+ \in \mathcal{L}_1 \cap \mathrm{supp}^+(F)$ and $X^- \in \mathcal{L}_1 \cap \mathrm{supp}^-(F)$. Then, with analogous arguments as in Section 4, we prove that $F^+ \in \mathcal{F}_\mathcal{L}(4)$, resulting in the sharp bound for rank-$(2,2)$-maxout networks.

If the positive support of a function $F \in \mathcal{F}_\mathcal{L}(2) \cap \mathcal{C}_\mathcal{L}$ is contained in the levels $\mathcal{L}_2$ and $\mathcal{L}_3$, then for every $S \in \mathrm{supp}^+(F) \cap \mathcal{L}_2$ there must be a $T \in \mathrm{supp}^+(F) \cap \mathcal{L}_3$ such that $T \supseteq S$ and $F(S) \leq F(T)$ since $\langle \alpha_{S,Y}, F \rangle = 0$. Applying the same argument to $T$, we conclude that $F(S) = F(T)$ and that there are no further subsets in $\mathrm{supp}^+(F)$ that are comparable to $S$ or $T$. Thus, we can match the subsets $S \in \mathcal{L}_2$ with the subsets $T \in \mathcal{L}_3$ such that $F(S) = F(T)$ and hence it follows that $\langle \alpha_{X,Y}, F^+ \rangle = \sum_{S \in \mathcal{L}_2} F^+(S) - \sum_{T \in \mathcal{L}_3} F^+(T) = 0$. By symmetry, the same holds if $\mathrm{supp}^-(F) \subseteq \mathcal{L}_2 \cup \mathcal{L}_3$. See Figure 4 for an illustration. Following this idea, we state the lemma for a more general case.

**Lemma 5.1.** *Let $\mathcal{L} = [X, Y]$ be a lattice of rank $n$ and $F \in \mathcal{F}_\mathcal{L}(k) \cap \mathcal{C}_\mathcal{L}$ with $n \geq 2k + 1$. If there are $i, j \in [n]_0$ such that $\mathrm{supp}^+(F) \subseteq \mathcal{L}_i \cup \mathcal{L}_j$ or $\mathrm{supp}^-(F) \subseteq \mathcal{L}_i \cup \mathcal{L}_j$, then it holds that $F^+ \in \mathcal{F}_\mathcal{L}(n-1)$.*

If there is a $X^+ \in \mathcal{L}_1 \cap \mathrm{supp}^+(F)$ and a $X^- \in \mathcal{L}_4 \cap \mathrm{supp}^-(F)$, then it holds that $\langle \alpha_{X,Y}, F^+ \rangle = \langle \alpha_{X^+,Y}, F \rangle = 0$ (Figure 4 and Lemma B.1 in the appendix). Thus we can assume that there are $X^+ \in \mathcal{L}_1 \cap \mathrm{supp}^+(F)$ and $X^- \in \mathcal{L}_1 \cap \mathrm{supp}^-(F)$. By proceeding analogously as in Section 4, we prove the following theorem.

**Theorem 5.2.** *It holds that $\mathcal{M}^2_{\mathcal{B}_d}(2) = \mathcal{V}_{\mathcal{B}_d}(4)$. In particular, the function $x \mapsto \{0, x_1, \ldots, x_4\}$ is not computable by a $\mathcal{B}^0_d$-conforming ReLU neural network with 2 hidden layers.*

## 6 The unimaginable power of maxouts

By Proposition 3.2, all functions in $\mathcal{V}_{\mathcal{B}_d}(\prod_{i=1}^{\ell} r_i)$ are representable by a $\mathcal{B}_d$-conforming rank-**r**-maxout network. In Section 5, we have seen that this bound is tight for the rank vector $(2, 2)$. In this section, we prove that this bound in general is not tight by demonstrating that the function $x \mapsto \{0, x_1, \ldots, x_6\}$ is computable by a $\mathcal{B}^0_d$-conforming rank-$(3, 2)$-maxout network.

**Proposition 6.1.** *Let $f_1, f_2 \in \mathcal{V}_{\mathcal{B}_7}(3)$ be the functions given by*

$$f_1 = 2 \cdot \sigma_{\{1,2\}} + \sigma_{\{1,4,5\}} + \sigma_{\{1,6,7\}} + \sigma_{\{2,4,6\}} + \sigma_{\{2,5,7\}}$$
$$f_2 = \sigma_{\{3,4,5\}} + \sigma_{\{3,6,7\}} + \sigma_{\{1,2,4\}} + \sigma_{\{1,2,5\}} + \sigma_{\{1,2,6\}} + \sigma_{\{1,2,7\}}$$

*Then it holds that $\max\{f_1, f_2\} \in \mathcal{V}_{\mathcal{B}_7}(7) \setminus \mathcal{V}_{\mathcal{B}_7}(6)$.*

*Proof Sketch.* Let $F_1 = \Phi(f_1)$ and $F_2 = \Phi(f_2)$. We write $i_1 \cdots i_n$ for $\{i_1, \ldots, i_n\}$ and $\overline{i_1 \cdots i_n}$ for $[7] \setminus \{i_1, \ldots, i_n\}$ and note that the sublattices $[12, \overline{3}], [13, \overline{2}], [23, \overline{1}], [3, \overline{12}], [2, \overline{13}], [1, \overline{23}], [\emptyset, \overline{123}], [123, [7]]$ form a partition of $[\emptyset, [7]]$.

We first show that on any of the above sublattices except $[1, \overline{23}]$, either $F_1$ or $F_2$ attains the maximum on all elements of the sublattice and that for $F := F_1 - F_2$ it holds that $\mathrm{supp}^+(F) \subseteq [1, \overline{23}] \cup 146 \cup 167$ and $\langle \alpha_{[\emptyset, [7]]}, F^+ \rangle = \langle \alpha_{12, \overline{3}}, F \rangle - F(146) - F(167) = -2$ and thus $F^+ \in \mathcal{F}_{\mathcal{L}} \setminus \mathcal{F}_{\mathcal{L}}(6)$. Then by looking at the partition into sublattices, we argue that $F \in \mathcal{C}_{\mathcal{L}}$ and thus by Lemma 3.3, we conclude that $\max\{f_1, f_2\} \in \mathcal{V}_{\mathcal{B}_7} \setminus \mathcal{V}_{\mathcal{B}_7}(6)$. $\square$

Hence $\max\{f_1, f_2\} = \sum_{M \subseteq [7]} \lambda_m \sigma_M$ with $\lambda_{[7]} \neq 0$ and since all functions in $\mathcal{V}_{\mathcal{B}_d}(6)$ are computable by a rank-$(3, 2)$-maxout network, we conclude that $x \mapsto \{x_1, \ldots, x_7\}$ is computable by a rank-$(3, 2)$-maxout network or equivalently:

**Theorem 6.2.** *The function $x \mapsto \{0, x_1, \ldots, x_6\}$ is computable by a rank-$(3, 2)$-maxout network.*

**Remark 6.3.** *One can check (e.g., with a computer) that $x \mapsto \{0, x_1, \ldots, x_6\}$ is computable by a rank-$(3, 2)$-maxout network with integral weights. This is particularly interesting in light of Haase et al. [2023], who prove a $\lceil \log_2(d + 1) \rceil$ lower bound for the case of integral weights and ReLU networks.*

## 7 Conclusion and Limitations

Characterizing the set of functions that a ReLU network with a fixed number of layers can compute remains an open problem. We established a doubly-logarithmic lower bound under the assumption that breakpoints lie on the braid fan. This assumption allowed us to exploit specific combinatorial properties of the braid arrangement. In the specific case of four dimensions, we reprove the tight bound for $\mathcal{B}^0_d$-conforming networks of Hertrich et al. [2023] with combinatorial arguments. Given that Bakaev et al. [2025b] showed that one can compute the maximum of 5 numbers with 2-layers, this implies that considering $\mathcal{B}^0_d$-conforming networks is a real restriction. While this indicates that the doubly-logarithmic lower bound may not extend to all networks, our approach provides a foundation for adapting these techniques toward more general depth lower bounds, for example, by looking at different underlying fans instead of just the braid fan.

**Acknowledgments** Moritz Grillo was supported by the Deutsche Forschungsgemeinschaft (DFG, German Research Foundation) — project 464109215 within the priority programme SPP 2298 "Theoretical Foundations of Deep Learning," and by Germany's Excellence Strategy — MATH+: The Berlin Mathematics Research Center (EXC-2046/1, project ID: 390685689). Part of this work

was completed while Christoph Hertrich was affiliated with Université Libre de Bruxelles, Belgium, and received support by the European Union's Horizon Europe research and innovation program under the Marie Skłodowska-Curie grant agreement No 101153187—NeurExCo.

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
