# OpenReview forum: "Depth-Bounds for Neural Networks via the Braid Arrangement"
_NeurIPS.cc/2025/Conference — NeurIPS 2025 oral_

### Official Review · Reviewer_AdUU · 2025-06-30

**Clarity:** 2
**Significance:** 3
**Originality:** 3
**Rating:** 5
**Confidence:** 3

**Summary:**

This work studies the expressive power of multi-layer perceptrons with ReLU activation function. They show a lower bound on the depth $l$ required to exactly express the maximum function in dimension $d$, ie the function $max(x_1, ...x_d)$. Their main result establishes that the depth of such network must be at least $l \geq \lceil\log(\log(d))\rceil$. Additionally, they show that a natural generalization of a conjecture on the expressive power of ReLU networks to any rank of maxout network is not true by providing a counter example for the max of 7 numbers.

**Questions:**

- This work focuses on exact representation of the target function. Have you thought about a similar lower bound on the depth when allowing for some approximation error with the target?
- At lines 107-109 you state that you disprove the conjecture with a counter example in dim 6. Do you think that the conjecture could still hold asymptotically with the same bound up to constants?
- It would be valuable to add a conclusion section to wrap-up your results.
- Finally, I noticed some lines which I think may contain typos or unclear sentence:

-- line 67: what does "integral" weights" mean?

-- line 89-90: could you link to the lemma or proposition that proves the "observation is that the set of functions that are representable by a B0d-conforming network forms a finite-dimensional vector space"?

-- line 110-112 Does it constitues a matching upper-bound to your lower bound? If yes, it could be nice to highlight it.

-- line 165: "fan" was not defined?

-- line 179: do you denote $F = \Phi \circ f$ here? If yes it would be good to add it.

-- line 190: Do you denote $[n]_0 = [n] \bigcup \{0\}$?

-- line 194: what is the * in $(R^L)^*$?

-- line 199: if I understand well $\sigma_M$ is a function, and hence should be $F(S)$. But don't we expect $F(S)$ to be scalar?

-- line 226: should you add $S_n\subset [d]$?

**Ethical Concerns:**

["NO or VERY MINOR ethics concerns only"]

**Final Justification:**

I believe this work should be accepted: the main contribution of proving a lower bound on the depth of a ReLU network to approximate the max function is new and a good contribution. Regarding the absence of experiments raised by a reviewer, I believe that this work is theoreticaI and does not require experiments, especially given the nature of the results proved (requiring exact approximation is a hard requirement to impose in experiments I believe).

Note that I could not carefully check all math details in the proof, which is extremely heavy in notations. The authors explained some of them in the rebuttal and I believe that adding some explanation/intuition in the notations could strengthen the paper.

**Limitations:**

The authors discuss adequately the limitations of their work. This is a theoretical work, which does not seem to present obvious negative societal impact.

**Paper Formatting Concerns:**

The font used seems different than the usual Neurip format font.

**Quality:**

3

**Strengths And Weaknesses:**

**Strengths**
- The motivation of this work is clearly motivated: the expressive power of depth of ReLU networks is not fully understood yet and subject to numerous ongoing works.
- The contributions are new and interesting: the main result provides a new non-trivial depth lower bound for expressing any piecewise linear function using ReLU networks.
- The paper writing is clear and formal, and I did not notice any major flaw.

**Weaknesses**
- This paper is *extremely* heavy in notations which makes it very hard to follow. I acknowledge that the notations are overall clearly defined and I do not immediately know how to simplify the notations but it really makes the reading hard currently.
- The setting of this work is quite restrictive: it applies to a single activation function and focuses on a setting which requires *exact* representation of the target function.

---

> ### Author Rebuttal · Authors · 2025-07-30
>
> We thank the reviewer for carefully reading and evaluation our paper and for the positive and encouraging feedback.
>
> > This paper is extremely heavy in notations which makes it very hard to follow. I acknowledge that the notations are overall clearly defined and I do not immediately know how to simplify the notations but it really makes the reading hard currently.
>
> Thanks for sharing this impression. We are happy to implement any concrete suggestion on how to improve this issue, including your comments below.
>
> > The setting of this work is quite restrictive: it applies to a single activation function and focuses on a setting which requires exact representation of the target function.
>
> Our results can be transferred to any activation function that is piecewise linear (including ReLU, leaky ReLU, maxout, ...). You are correct in that they don't apply to smooth activation functions. In our current version, our results do not imply anything for approximation, but we see the exact representations as a necessary base case, that is already substantially non-trivial.
>
> > This work focuses on exact representation of the target function. Have you thought about a similar lower bound on the depth when allowing for some approximation error with the target?
>
> See above.
>
> > At lines 107-109 you state that you disprove the conjecture with a counter example in dim 6. Do you think that the conjecture could still hold asymptotically with the same bound up to constants?
>
> It could be either way, we do not want to make a guess at this point.
>
> > It would be valuable to add a conclusion section to wrap-up your results.
>
> We will do this, see also our discussion with Reviewer jBBb.
>
> > line 67: what does "integral" weights" mean?
>
> It means that all weights are in $\mathbb{Z}$.
>
> > line 89-90: could you link to the lemma or proposition that proves the "observation is that the set of functions that are representable by a B0d-conforming network forms a finite-dimensional vector space"?
>
> This is implied by Prop. 2.2. We are happy to add such a reference.
>
> > line 110-112 Does it constitues a matching upper-bound to your lower bound? If yes, it could be nice to highlight it.
>
> Unfortunately, the problem setups are slightly different, so while morally the two bounds match, this is not a formal match.
>
> > line 165: "fan" was not defined?
>
> We will add a definition: A fan is a polyhedral complex in which every polyhedron is a cone anchored at the origin.
>
> > line 179: do you denote $F=\Phi\circ f$ here? If yes it would be good to add it.
>
> Not exactly: the meaning is that $\Phi(f)$ is defined as the set function $F$ that sends $S$ to $f(1_S)$. We will make this clearer in the final version.
>
> > line 190: Do you denote $[n]_0 = [n]\cup 0$?
>
> Yes
>
> > line 194: what is the * in $(R^L)^*$?
>
> It is a standard notation for the dual space, the set of linear functions mapping from $R^L$ to $R$. We will add an explanation.
>
> > line 199: if I understand well $\sigma_M$ is a function, and hence should be $F(S)$. But don't we expect $F(S)$ to be scalar?
>
> This is indeed a typo. $F(S)$ is a scalar, as you say. The formula should not contain the $\sigma_M$, it should just be a linear combination of the $\lambda$s.
>
> > line 226: should you add $S_n\subset[d]?
>
> Yes, this can be added here.
>
> > Paper Formatting Concerns: The font used seems different than the usual Neurip format font.
>
> You are correct and we are sorry about this. By accident, we loaded the package lmodern, which changes the font. But our font takes up more space than the usual NeurIPS font, so we do not overshoot the page limit when changing it back to the NeurIPS font. In contrast, we will even have a little bit of additional space for the final version.

---

> > ### Comment · Reviewer_AdUU · 2025-08-04
> >
> > I thank the authors for their reply to my questions. In light of their answer and the answer to other reviewers questions, and even if I am not very familiar with the mathematical tools used and could not verify the entirety of the results, I am comforted in that this work should be accepted. I will increase my score accordingly.
> >
> > I would just like to remark that I respectfully disagree with their reply *"we see the exact representations as a necessary base case, that is already substantially non-trivial."*. I have the opinion that imposing exact representation is a very strong requirement that does not necessarily reflects practical scenario. This is evidenced by the fact that the results can not be transfered to smooth activations. Imposing exact approximation makes the theory very rigid, and does not give many insight on practical scenarios, or does not generalize well. This does not change that the combinatorial results proved here are still very interesting and constitute a good contribution.

---

### Official Review · Reviewer_7cCK · 2025-06-30

**Clarity:** 3
**Significance:** 3
**Originality:** 3
**Rating:** 5
**Confidence:** 4

**Summary:**

This paper studies the theoretical question of how many layers a deep ReLU network requires to exactly represent any continuous piecewise linear function in d dimensions. The conjectured lower bound is log(d) and the present paper proves a conditional lower bound of log(log(d)). Specifically, the authors show that if the outputs of each neuron in the network are piecewise linear functions commensurate with a certain lattice, then the network requires a depth of at least log(log(d)) to represent the maximum of d numbers.

The authors also give a combinatorial proof a previous result which was proved using a large computed calculation and, interestingly, show that maxout networks don't satisfy the analogous depth lower bound.

**Questions:**

I think that there is a typo on line 70. It seems that the required depth of the network grows as $N\rightarrow \infty$, which doesn't make sense since the restriction on the weights is now weaker. Should the $\log N$ be in the denominator?

**Ethical Concerns:**

["NO or VERY MINOR ethics concerns only"]

**Final Justification:**

This is a technically interesting and solid paper. The authors have answered all of my questions and I have no concerns. I stick with my original assessment of accept.

**Limitations:**

yes

**Paper Formatting Concerns:**

No paper formatting concerns.

**Quality:**

3

**Strengths And Weaknesses:**

Strengths: The paper addresses a very interesting theoretical question, namely the minimal depth required for deep ReLU networks to exactly represent and CPWL function. Despite its simple statement, this problem has proved to be surprisingly technically challenging. The authors make very interesting partial progress toward this problem despite its apparent difficulty. In addition, the results on general maxout networks are highly interesting and suggest that something is special about the ReLU network problem.

Weaknesses: The main theorem proved in the paper is conditional on the structure of the neural network, so a complete solution to the desired problem is still missing. However, the exceptional difficulty of this combinatorial problem makes this perhaps inevitable.

---

> ### Author Rebuttal · Authors · 2025-07-30
>
> We thank the reviewer for carefully reading and evaluation our paper and for the positive and encouraging feedback.
>
> > Weaknesses: The main theorem proved in the paper is conditional on the structure of the neural network, so a complete solution to the desired problem is still missing. However, the exceptional difficulty of this combinatorial problem makes this perhaps inevitable.
>
> We agree with your estimation of the situation.
>
> > I think that there is a typo on line 70. It seems that the required depth of the network grows as $N\rightarrow \infty$, which doesn't make sense since the restriction on the weights is now weaker. Should the $\log N$ be in the denominator?
>
> Yes, this is a typo, d and N should be exchanged in the expression. Thanks for pointing it out.

---

> > ### Comment · Reviewer_7cCK · 2025-08-05
> >
> > I thank the authors for their response. I don't have any further questions or concerns.

---

### Official Review · Reviewer_jBBb · 2025-07-02

**Clarity:** 3
**Significance:** 2
**Originality:** 3
**Rating:** 4
**Confidence:** 2

**Summary:**

This paper studies the problem of determining minimal depth required for representing continuous piecewise linear (CPWL) functions exactly using ReLU neural networks. Specifically, it aims to resolve an open problem regarding the number of hidden layers needed for exact representation. The authors focus on neural networks conforming to specific polyhedral complexes, the braid arrangement, and provide novel theoretical results. The authors prove that $\Omega(\log \log d)$ layers are required to represent the function $\max\{0,x_1,\ldots,x_d\}$. This is the first non-constant lower bound in this setting without weight restrictions. It also provides a combinatorial proof that two hidden layers are insufficient to compute  $\max\{0,x_1,\ldots,x_4\}$ under this model, and shows that a rank-(3,2) maxout network can compute $\max\{0,x_1,\ldots,x_6\}$, disproving a natural upper bound based on rank multiplication.

**Questions:**

1.Could one empirically confirm that a ReLU network with $l < \log \log d$ layers fails to approximate $\max\{0, x₁,...,x_d\}$? How would you test this?
2.Are there concrete examples where the Ω(log log d) lower bound is actually tighter for specific values of d or other specific cases?
3.Is it possible to prove more generally that for certain rank vectors,  the expressivity of maxout networks indeed exceeds the bound given by $\prod_i r_i$?

**Ethical Concerns:**

["NO or VERY MINOR ethics concerns only"]

**Final Justification:**

I have no other concerns.

**Limitations:**

Suggestions:
This paper presents reliable theoretical analysis and establishes a novel lower bound related to the depth on the expressivity of ReLU networks within specific conditions. However, it lacks empirical verification, even on simple network architectures such as fully connected networks, which limits its significance. As such, it is difficult to consider it a theoretical contribution with substantial impact.

If the authors could provide experimental results to support the $\omega(\log \log d)$ lower bound in a future revision, the paper would have strong potential for acceptance. Moreover, while the definition of $B_d$-conforming networks is theoretically well-defined and precise, the main claims would be significantly strengthened if the authors could empirically demonstrate—or at least provide some examples—showing that real-world neural networks tend to exhibit such structure, either inherently or approximately. Establishing this connection would enhance the practical relevance of the proposed lower bounds and make the results more compelling to a broader audience. Otherwise, the current submission resembles a combinatorial-algebraic exploration whose contribution falls short of the standards for acceptance at NeurIPS.

Finally, although the main results of this paper are clearly presented, I could not find a summary or discussion of the work, particularly regarding its limitations, despite the authors' claim in the checklist that such limitations and conclusions are discussed. If I have misunderstood or overlooked this, I would appreciate clarification.

**Paper Formatting Concerns:**

not formatting issues.

**Quality:**

2

**Strengths And Weaknesses:**

Strength:

The $\Omega(\log \log d)$ lower bound improves the previous constant-depth bounds.
This paper presents a clear description of its results with strict and reliable analytical techniques.

Weakness:
Absence of empirical verification on its main results (even on vanilla examples of fully connected networks, see Suggestions).

---

> ### Author Rebuttal · Authors · 2025-07-30
>
> We thank the reviewer for carefully reading and evaluation our paper and for the constructive feedback.
>
> > Weakness: Absence of empirical verification on its main results (even on vanilla examples of fully connected networks, see Suggestions).
>
> Our work is fundamental research on the expressive power of ReLU networks, and as such of theoretical nature. Many related previous papers published at top-tier ML conferences are purely theoretical papers, including the cited papers by Hertrich et al. 2023; Haase et al. 2023; Averkov et al. 2025. We agree that an empirical verification would be a nice add-on, but designing it in an insightful way would be nontrivial and is beyond the scope of our paper.
>
> > 1. Could one empirically confirm that a ReLU network with l < log log d layers fails to approximate max{0,x_1,...,x_d}? How would you test this?
>
> See response above.
>
> > 2. Are there concrete examples where the Ω(log log d) lower bound is actually tighter for specific values of d or other specific cases?
>
> We are not aware of tight examples. The best upper bound is singly-logarithmic in d. Nevertheless, our lower bound is the first proof of a non-constant lower bound, and as such a big step towards matching upper and lower bounds eventually.
>
> > 3. Is it possible to prove more generally that for certain rank vectors, the expressivity of maxout networks indeed exceeds the bound given by $\prod_i r_i$?
>
> Our result can be stacked: by alternating $\ell$ rank-2 and $\ell$ rank-3 layers, one can compute the maximum of $7^\ell$ numbers. We are, however, not aware of a more general construction of this type.
>
> > This paper presents reliable theoretical analysis and establishes a novel lower bound related to the depth on the expressivity of ReLU networks within specific conditions. However, it lacks empirical verification, even on simple network architectures such as fully connected networks, which limits its significance. As such, it is difficult to consider it a theoretical contribution with substantial impact. If the authors could provide experimental results to support the $\omega(\log \log d)$ lower bound in a future revision, the paper would have strong potential for acceptance. Moreover, while the definition of B_d-conforming networks is theoretically well-defined and precise, the main claims would be significantly strengthened if the authors could empirically demonstrate—or at least provide some examples—showing that real-world neural networks tend to exhibit such structure, either inherently or approximately. Establishing this connection would enhance the practical relevance of the proposed lower bounds and make the results more compelling to a broader audience. Otherwise, the current submission resembles a combinatorial-algebraic exploration whose contribution falls short of the standards for acceptance at NeurIPS.
>
> See above. We kindly disagree and think that the theoretical contribution alone is a significant contribution to the NeurIPS audience.
>
> > Finally, although the main results of this paper are clearly presented, I could not find a summary or discussion of the work, particularly regarding its limitations, despite the authors' claim in the checklist that such limitations and conclusions are discussed. If I have misunderstood or overlooked this, I would appreciate clarification.
>
> Thanks for pointing this out. While we think that all assumptions, implications, and limitations are appropriately discussed around the theorem statements in the introduction, we are happy to include a specific paragraph summarizing these points. In such a paragraph we are also happy to discuss potential approaches for empirical verification, as you suggest.

---

> > ### Comment · Reviewer_jBBb · 2025-08-09
> >
> > Dear Authors,
> >
> > Thank you for your responses. I believe this is a problem of significance, but the submission has not made  a good presentation on the significance. From the information provided in the paper so far, we can only outline the following
> >
> > 1. It is an open problem established a few years ago
> > 2. There have been partial results on the conjecture
> > 3. This paper has improved the current result/achieved the best result, for the first time.
> >
> > But these are default elements of a theoretical paper, and having these elements does not automatically make the result significant.
> > How was this problem practically motivated in the first place?
> > Why the resolvent of the problem still matters beyond purely mathematical challenges when years have passed?
> >
> > Clarifying these points does not go against the theoretial nature. In fact the author could have just cited some contents that indicate knowing the complexity bound in expressing piecewise linear functions is practically useful.
> >
> > Otherwise, if even authors cannot address the practical motivation, how could someone who missed the intensive development of deep learning theory understand the significance.  There should be a difference in evaluating results that are 1. theoretically interesting but practically useless (at least in a short period); 2. theoretically less strong but practically impactful; 3. theoretically strong and practically useful. The write-up has not made it easy for one to map the contribution to a proper category. And the response seems having mis-understood (possibly due to my presentation) my original point.
> >
> > I have no problem for acceptance of the paper, but there are probably  only handful of problems that do not need proper motivations for readers to know about the importance. Many theoretical problems in computer science have practical origins in the first place. I still suggest authors to consider adding some comments on how these results are (or are not) practically useful.

---

### Official Review · Reviewer_LfjK · 2025-07-05

**Clarity:** 3
**Significance:** 3
**Originality:** 3
**Rating:** 5
**Confidence:** 1

**Summary:**

The paper tackles the questionn of depth lower bounds for ReLU networks that are conforming with braid fan and gave a lower bound on how many ReLU hidden layers needed to exactly represent the maximum of d number in such a network: log log d.

**Questions:**

I do not have any questions.

**Ethical Concerns:**

["NO or VERY MINOR ethics concerns only"]

**Final Justification:**

I didn't have any concerns with this paper and keep my rating as is.

**Limitations:**

Yes

**Quality:**

3

**Strengths And Weaknesses:**

As someone who is outside this field, I appreciate the motivation of the paper. Using lattice-theoretic arguments to derive a log log lower bound is elegant, even if most of the proof machinery was beyond what I can comfortably follow without spending more time on the background.

I also found the maxout network section quite interesting. The fact that the natural generalization of the upper bound (based on the product of ranks) doesn't hold in general and that (3,2)-maxout can already represent max of 7 variables  was unexpected. It's a neat contrast to the tightness results for (2,2)-ranked networks.

That said, this is pretty far from the type of work I usually engage with, so I don’t have the right context to assess novelty nor significance within this area. But the writing was clear enough to follow the high-level structure, and the results seem carefully stated with appropriate proofs.

---

> ### Author Rebuttal · Authors · 2025-07-30
>
> We thank the reviewer for carefully reading and evaluation our paper and for the positive and encouraging feedback.

---

### Note · Authors · 2025-08-13

Dear AC,

We sincerely thank all reviewers for their time and their detailed, constructive feedback. We are very encouraged that the majority of reviewers found our work to be a strong contribution.

A point of discussion, raised by Reviewer jBBb, was the need to better articulate the broader significance of our work. We appreciate this push for clarity. To address this, we will add a dedicated discussion section to the final manuscript.

This section will frame our work by explaining that while we are not aware of immediate "plug-and-play" applications, our research provides a fundamental contribution to understanding deep neural networks. We believe NeurIPS is a vital venue for such theoretical work, a view exemplified by many of the foundational papers we cite.

In this regard, we will highlight two key aspects of our work's significance:
- It provides theoretical justification for common architectural choices. Proving depth lower bounds on computing the max function helps formally explain why elements like max-pooling layers are powerful and cannot be easily replaced by shallow stacks of standard ReLU layers.
- It advances a foundational research direction with deep connections to other fields such as polyhedral and tropical geometry. Studying exact representation, as we do, is a cornerstone for understanding expressive power. This approach has led to crucial insights for algorithmic tasks like network verification, training and their complexity analysis, as noted in our introduction (lines 25-31).

We believe this added context will make the paper's impact clear to the readers, reinforcing the positive assessment from the other reviewers who agreed that they "appreciate the motivation," that "the paper addresses a very interesting theoretical question," and that "this work is clearly motivated."

Best regards,

The authors

---

### Decision · Program_Chairs · 2025-09-17

**Decision:**

Accept (oral)

**Comment:**

The reviewers were overwhelmingly positive, with all four reviewers recommending acceptance. The paper is technically solid, clean, clearly written, and addresses a highly relevant theoretical question. Since the result is neat and closes a research path, I'd recommend the paper for an oral.
The minor issues raised by the reviewers, such as typos and notation clarity, were addressed by the authors and do not detract from the overall quality of the work. For these reasons, I recommend acceptance.